# Pixel-Level Cycle Association: A New Perspective for Domain Adaptive Semantic Segmentation

**Guoliang Kang**[1], **Yunchao Wei**[2],*, **Yi Yang**[2], **Yueting Zhuang**[3], **Alexander G. Hauptmann**[1]

[1] School of Computer Science, Carnegie Mellon University
[2] ReLER, University of Technology Sydney [3] Zhejiang University
kgl.prml@gmail.com, alex@cs.cmu.edu
{yunchao.wei, yi.yang}@uts.edu.au
yzhuang@zju.edu.cn

## Abstract

Domain adaptive semantic segmentation aims to train a model performing satisfactory pixel-level predictions on the *target* with only out-of-domain (*source*) annotations. The conventional solution to this task is to minimize the discrepancy between source and target to enable effective knowledge transfer. Previous domain discrepancy minimization methods are mainly based on the adversarial training. They tend to consider the domain discrepancy globally, which ignore the pixel-wise relationships and are less discriminative. In this paper, we propose to build the pixel-level cycle association between source and target pixel pairs and contrastively strengthen their connections to diminish the domain gap and make the features more discriminative. To the best of our knowledge, this is a new perspective for tackling such a challenging task. Experiment results on two representative domain adaptation benchmarks, *i.e.* GTAV $\rightarrow$ Cityscapes and SYNTHIA $\rightarrow$ Cityscapes, verify the effectiveness of our proposed method and demonstrate that our method performs favorably against previous state-of-the-arts. Our method can be trained end-to-end in one stage and introduces no additional parameters, which is expected to serve as a general framework and help ease future research in domain adaptive semantic segmentation. Code is available at https://github.com/kgl-prml/Pixel-Level-Cycle-Association.

## 1 Introduction

Semantic segmentation is a challenging task because it requires pixel-wise understandings of the image which is highly structured. Recent years have witnessed the huge advancement in this area, mainly due to the rising of deep neural networks. However, without massive pixel-level annotations, to train a satisfactory segmentation network remains challenging. In this paper, we deal with the semantic segmentation in the domain adaptation setting which aims to make accurate pixel-level predictions on the target domain with only out-of-domain (source) annotations.

The key to the domain adaptive semantic segmentation is how to make the knowledge learned from the distinct source domain better transferred to the target. Most previous works employ adversarial training to minimize the domain gap existing in the image [11, 12, 47] or the representations [41, 31, 44] or both [19, 49]. Most adversarial training based methods either focus on the discrepancy globally or treat the pixels from both domains independently. All these methods ignore the abundant in-domain and cross-domain pixel-wise relationships and are less discriminative. Recently, many works resort to self-training [48, 27, 54] to boost the segmentation performance. Although self-training leads to promising improvement, it largely relies on a good initialization, is hard to tune and

---

usually leads to a result with large variance. Our work focuses on explicitly minimizing the domain discrepancy and is orthogonal to the self-training based method.

In this paper, we provide a new perspective to diminish the domain gap via exploiting the pixel-wise similarities across domains. We build the cross-domain pixel association cyclically and draw the associated pixel pairs closer compared to the unassociated ones. Specifically, we randomly sample a pair of images (*i.e.* a source image and a target image). Then starting from each source pixel, we choose the most similar target pixel and in turn find its most similar source one, based on their features. The cycle-consistency is satisfied when the starting and the end source pixels come from the same category. The associations of the source-target pairs which satisfy the cycle-consistency are contrastively strengthened compared to other possible connections. Because of the domain gap and the association policy tending to choose the easy target pixels, the associated target pixels may only occupy a small portion of the whole image/feature map. In order to provide guidance for all of the target pixels, we perform spatial feature aggregation for each target pixel and adopt the internal pixel-wise similarities to determine the importance of features from other pixels. In this way, the gradients with respect to the associated target pixels can also propagate to the unassociated ones.

We verify our method on two representative benchmarks, *i.e.* GTAV $\rightarrow$ Cityscapes and SYNTHIA $\rightarrow$ CityScapes. With mean Intersection-over-Union (mIoU) as the evaluation metric, our method achieves more than 10% improvement compared to the network trained with source data only and performs favorably against previous state-of-the-arts.

In a nutshell, our contributions can be summarized as: 1) We provide a new perspective to diminish the domain shift via exploiting the abundant pixel-wise similarities. 2) We propose to build the cross-domain pixel-level association cyclically and contrastively strengthen their connections. Moreover, the gradient can be diffused to the whole target image based on the internal pixel-wise similarities. 3) We demonstrate the effectiveness of our method on two representative benchmarks.

## 2    Related Work

**Domain Adaptation.** Domain Adaptation has been studied for decades [3, 2, 13, 35, 17, 6, 30] in theory and in various applications. Recent years, deep learning based domain adaptation methods significantly improve the adaptation performance. Plenty of works have been proposed to mitigate the domain gap either from the image level, *e.g.* adversarial training based image-to-image translation [5, 19, 32, 23], or from the representation level [38, 16, 28, 43, 29, 22, 18], *e.g.* a series of works based on adversarial training [16, 43] or Maximum Mean Discrepancy (MMD) [28, 29, 22], *etc.*.

**Adversarial training based adaptive segmentation.** Plenty of works fall into this category. Among these works, adversarial training is employed mainly in two ways: 1) Acting as a kind of data augmentation via style transfer across domains [11, 12, 7, 47, 49, 24]. It can be viewed as mitigating the domain gap from the image level. For example, DISE [7] disentangled the image into domain-invariant structure and domain-specific texture representations in realizing image-translation across domains. 2) Making the features or the network predictions indistinguishable across domains [41, 42, 36, 20, 21, 31, 44, 15, 34, 46]. For example, Tsai *et al.* [41, 42] proposed applying adversarial training to the multi-level network predictions to mitigate the domain shift. And some works combined two kinds of adversarial training together, *e.g.* [19, 49].

**Self-training based adaptive segmentation.** Self-training based methods tend to iteratively improve the segmentation performance [48, 27, 54, 37, 25]. The PyCDA [27] constructed the pyramid curriculum to provide various properties about the target domain to guide the training. Chen *et al.* [9] proposed using maximum squares loss and multi-level self-produced guidance to train the network. The CRST [54] proposed to use the soft pseudo-labels and smooth network predictions to make self-training less sensitive to the noisy pseudo-labels. Some works [26, 51, 24] resort to self-training as a second stage training to further boost their adaptation performance. However, self-training has some issues in nature. It relies on good initialization, which is usually pretrained on source or adapted on target via other method beforehand. It remains unknown that to what extent should we pretrain our model. Also, the training is sensitive to the noise and easy to collapse.

In this paper, we mainly focus on how to explicitly utilize the cross-domain pixel-wise relationships to mitigate the domain gap, which has been ignored by most previous works. Our method performs one

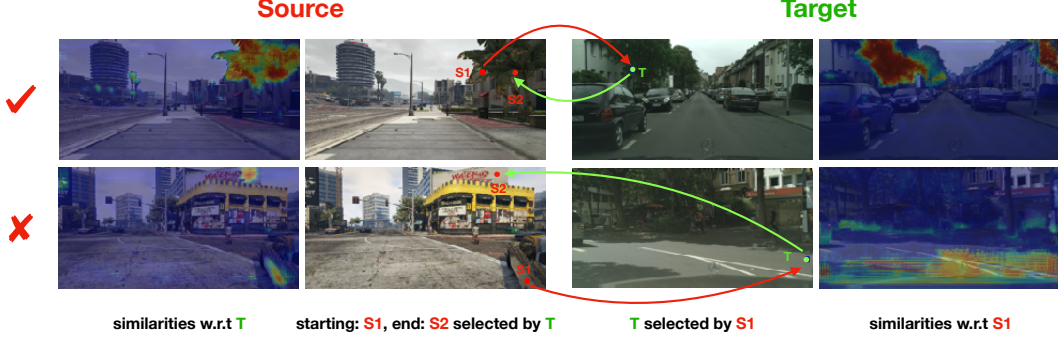

| | | | |
|---|---|---|---|
| **Source** | | **Target** | |

similarities w.r.t T     starting: S1, end: S2 selected by T     T selected by S1     similarities w.r.t S1

Figure 1: Illustration of pixel-level cycle association. In the figure, the red and green points denote the pixels in source and target images respectively. Starting from the source pixel "S1", the target pixel "T", which has the highest similarity with "S1" among all the target pixels, is selected. In turn, with respect to "T", the most similar source pixel "S2" is selected. If the cycle consistency is satisfied, *i.e.* "S1" and "S2" come from the same category, as shown in the top row, the association is valid, otherwise the association is regarded as invalid, as shown in the bottom row.

stage end-to-end training, which is orthogonal to the self-training based methods, data augmentation methods via style transfer, *etc*.

# 3 Method

## 3.1 Background

In domain adaptive semantic segmentation, we have a set of labeled source data $S$ and unlabeled target data $T$. The underlying set of categories for the target are the same with the source. We aim to train a network $\Phi_\theta$ to classify each target pixel into one of the underlying $M$ classes. During training, the network $\Phi_\theta$ is trained with the labeled source data and unlabeled target data. We aim to mitigate the domain shift during training, via our proposed pixel-level cycle association (PLCA).

## 3.2 Pixel-Level Cycle Association

Cycle consistency has proved to be effective in various applications [53, 45, 52, 1]. In domain adaptation, CycleGAN [53], which is established upon the cycle consistency, is usually employed to transfer the style across domains to diminish the image-level domain shift. However, in semantic segmentation which requires pixel-wise understanding of the structured data, image-level correspondence cannot fully exploit abundant pixel-wise relationships to improve the pixel-wise discriminativeness.

In this paper, we propose to use the cycle consistency to build the pixel-level correspondence. Specifically, given the source feature map $F^s \in \mathbb{R}^{C \times H^s \times W^s}$ and the target feature map $F^t \in \mathbb{R}^{C \times H^t \times W^t}$, for an arbitrary source pixel $i \in \{0, 1, \cdots, H^s \times W^s - 1\}$ in $F^s$, we compute its pixel-wise similarities with all of the target pixels in $F^t$. We adopt cosine similarity between features to measure the pixel-wise similarity, *i.e.*

$$D(F_i^s, F_j^t) = \langle \frac{F_i^s}{\|F_i^s\|}, \frac{F_j^t}{\|F_j^t\|} \rangle \tag{1}$$

where $F_i^s \in \mathbb{R}^C$ and $F_j^t \in \mathbb{R}^C$ are features of source pixel $i$ and target pixel $j$ respectively. The $\langle \cdot \rangle$ denotes the inner-product operation, and $\|\cdot\|$ denotes the $L^2$-norm.

As shown in Fig. 1, for each source pixel $i$, we select the target pixel $j^*$ as

$$j^* = \underset{j' \in \{0, 1, \cdots, H^t \times W^t - 1\}}{\operatorname{argmax}} D(F_i^s, F_{j'}^t) \tag{2}$$

Similarly, given selected target pixel $j^*$, we could in turn select the source pixel $i^*$ with

$$i^* = \underset{i' \in \{0, 1, \cdots, H^s \times W^s - 1\}}{\operatorname{argmax}} D(F_{j^*}^t, F_{i'}^s) \tag{3}$$

The cycle-consistency is satisfied when the starting source pixel $i$ and the end source pixel $i^*$ come from the same semantic class, *i.e.* $y_i^s = y_{i*}^s$ where $y_i^s$ and $y_{i*}^s$ denote the ground-truth labels for source pixel $i$ and $i^*$ respectively. For the associated pairs $(i, j^*)$ and $(j^*, i^*)$ which satisfy the cycle-consistency, we choose to enhance both of their connections. A straightforward way is to directly maximize their feature similarities during training. However, besides the semantic information, pixel-level features contain abundant context and structural information. Directly maximizing their similarities may introduce bias because the associated pixel pairs come from two distinct images and their context or structure may not be exactly the same. Thus we choose to contrastively strengthen their connections, *i.e.* making the similarity of associated pixel pair higher than any other possible pixel pair. Specifically, we minimize the following loss during training, the form of which is similar to InfoNCE loss [33, 40] in the contrastive learning

$$\mathcal{L}^{fass} = -\frac{1}{|\hat{I}^s|} \sum_{i \in \hat{I}^s} \log\{\frac{\exp\{D(F_i^s, F_{j*}^t)\}}{\sum_{j'} \exp\{D(F_i^s, F_{j'}^t)\}} \frac{\exp\{D(F_{j*}^t, F_{i*}^s)\}}{\sum_{i'} \exp\{D(F_{j*}^t, F_{i'}^s)\}}\} \tag{4}$$

where $\hat{I}^s$ denotes the set of starting source pixels which are successfully associated with the target ones, and $|\hat{I}^s|$ is the number of such source pixels.

Moreover, we want to improve the contrast among the pixel-wise similarities to emphasize the most salient part. Thus we perform the following contrast normalization on $D$ before the softmax operation

$$D(F_i^s, F_{j'}^t) \leftarrow \frac{D(F_i^s, F_{j'}^t) - \mu_{s \to t}}{\sigma_{s \to t}}, D(F_{j*}^t, F_{i'}^s) \leftarrow \frac{D(F_{j*}^t, F_{i'}^s) - \mu_{t \to s}}{\sigma_{t \to s}} \tag{5}$$

where

$$\mu_{s \to t} = \frac{1}{H^t \times W^t} \sum_{j'} D(F_i^s, F_{j'}^t), \sigma_{s \to t} = \sqrt{\frac{1}{H^t \times W^t - 1} \sum_{j'} (D(F_i^s, F_{j'}^t) - \mu_{s \to t})^2}$$

$$\mu_{t \to s} = \frac{1}{H^s \times W^s} \sum_{i'} D(F_{j*}^t, F_{i'}^s), \sigma_{t \to s} = \sqrt{\frac{1}{H^s \times W^s - 1} \sum_{i'} (D(F_{j*}^t, F_{i'}^s) - \mu_{t \to s})^2} \tag{6}$$

Roughly speaking, Eq. (5) illustrates that for the network backward process, the incoming gradient with respect to $D$ will be adaptively adjusted by the contrast of relevant similarities, *i.e.*, scaled by the computed statistic $\sigma$. When the $\sigma$ is large, the gradient will be scaled down because it implies the contrast among relevant similarities is large, and vice versa. In such way, we are able to safely strengthen the cross-domain pixel-level similarities without introducing too much bias to the training.

### 3.3 Gradient Diffusion via Spatial Aggregation

Based on the proposed cycle association policy, target pixels can be selected to approach to the corresponding source pixels. However, usually only a small portion of target pixels could be covered via the association. The reason comes from several aspects. According to our association policy, there may exist duplicated target pixels associated in different cycles. Moreover, for a pair of randomly sampled source and target images, partial target pixels are semantically distinct from all of the source pixels and should not be associated in nature.

In order to regularize the training of all the target pixels, we spatially aggregate the features for each target pixel before performing the cycle association, *i.e.* the feature of each pixel is represented as a weighted sum of features from all the other target pixels in the same image,

$$\hat{F}_j^t = (1 - \alpha)F_j^t + \alpha \sum_{j'} w_{j'} F_{j'}^t \tag{7}$$

$$w_{j'} = \frac{\exp\{D(F_j^t, F_{j'}^t)\}}{\sum_{j'} \exp\{D(F_j^t, F_{j'}^t)\}} \tag{8}$$

where $\alpha \in [0, 1]$ is a constant that controls the ratio of aggregated features. Empirically we find that comparable results can be achieved by setting $\alpha \in [0.5, 1.0]$. And in our method, we set $\alpha = 0.5$. Note that we also perform contrast normalization on $D$ as illustrated in Eq. (6) before the softmax.

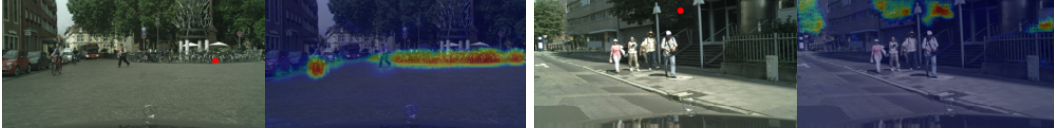

Figure 2: Illustration of internal similarities with respect to specific target pixels. With the spatial aggregation, the gradients can diffuse to the whole image via the seed pixels (*i.e.* the red points in target images), while emphasizing the similar areas (the highlighted part in the similarity map). Also, the features of the seed pixels can be strengthened by aggregating similar features of other pixels .

Then the aggregated target features $\hat{F}_j^t$ are adopted to perform the cycle association. During the backward process, the associated target pixel can be viewed as a "seed" which diffuses the gradient spatially to multiple "receivers" (*i.e.* all the other target pixels whose features are aggregated into the seed via Eq. (7)). Note that the gradient is diffused unevenly, the amount of which is determined by the similarity of features between the receiver and the seed, as shown in Fig. 2.

A side effect brought by Eq. (7) is that the target pixel-wise features are strengthened, which reduces the ratio of invalid associations and benefits the early stage of adaptation.

### 3.4 Multi-Level Association

Besides applying pixel-level cycle association to the last feature maps of the backbone, we also employ association on the final predictions. Such multi-level association may benefit the adaptation because the domain discrepancy of features cannot be perfectly eliminated, so that the following classifier may easily find shortcut to overfit to the source and degenerate the adaptation performance. Thus we propose to apply pixel-level cycle association to the final network predictions to reduce the risk of such overfitting.

We follow the same rule as discussed in Section 3.2 to perform the association. The difference is that we use the negative Kullback-Leibler (KL) divergence as the similarity measure, *i.e.*

$$D^{kl}(P_i^s, P_j^t) = -\sum_c P_i^s(c) \log \frac{P_i^s(c)}{P_j^t(c)}, \quad D^{kl}(P_j^t, P_i^s) = -\sum_c P_j^t(c) \log \frac{P_j^t(c)}{P_i^s(c)} \quad (9)$$

where $P \in \mathbb{R}^{M \times H \times W}$ denote the predicted probabilities for $M$ classes. And $P_i^s(c)$ and $P_j^t(c)$ denote the probability of source pixel $i$ and target pixel $j$ to be in class $c \in \{0, 1, \cdots, M-1\}$, respectively. Note that $D^{kl}(P_i^s, P_j^t)$ is different from $D^{kl}(P_j^t, P_i^s)$ because the direction of comparison matters in measuring the KL divergence. It is reasonable because when associating target pixel $j$ with source pixel $i$, we expect the one better matching with the mode of $P_i^s$ could be selected, and vice versa.

Similar to Eq. (5), the association loss on the predictions $P$ is,

$$\mathcal{L}^{cass} = -\frac{1}{|\hat{I}^s|} \sum_{i \in \hat{I}^s} \log\{ \frac{\exp\{D^{kl}(P_i^s, P_{j*}^t)\}}{\sum_{j'} \exp\{D^{kl}(P_i^s, P_{j'}^t)\}} \frac{\exp\{D^{kl}(P_{j*}^t, P_{i*}^s)\}}{\sum_{i'} \exp\{D^{kl}(P_{j*}^t, P_{i'}^s)\}} \} \quad (10)$$

We also apply contrast normalization like Eq. (6) to normalize the $D^{kl}$ before the softmax operation. Moreover, we also adopt spatial aggregation to enable the gradient to diffuse to the whole image.

By applying multi-level cycle association, in addition to the effect of reducing overfitting to the source, the adaptation may benefit from mining and exploiting multi-granular pixel-wise relationships. The multi-level associations are complementary to each other and both of them contribute to alleviating the domain shift.

### 3.5 Objective

During the adaptation, for the source data, we train it with the pixel-wise cross-entropy loss, *i.e.*

$$\mathcal{L}^{ce} = -\frac{1}{|I^s|} \sum_{i \in I^s} \log P_i^s(y_i^s) \quad (11)$$

where $I^s$ denotes the source image. Additionally, as the amount of pixels for different categories vary a lot, the lovász-softmax loss [4] $\mathcal{L}^{lov}$ is imposed on source data to alleviate the negative effect of imbalanced data. And through association, the target predictions can also be more balanced.

Meanwhile, we employ the multi-level association loss to mitigate the domain discrepancy, *i.e.*

$$\mathcal{L}^{asso} = \mathcal{L}^{fass} + \mathcal{L}^{cass} \tag{12}$$

As the predictions $P$ carry the semantic relationship information, we encourage softer predictions by adaptively imposing the Label Smooth Regularization (LSR) [39] on both source and target, *i.e.*

$$\mathcal{L}^{lsr} = -\frac{1}{M}\{\frac{1}{|I^s|}\sum_{i \in I^s}\gamma_i \sum_c \log P_i^s(c) + \frac{1}{|I^t|}\sum_{j \in I^t}\gamma_j \sum_c \log P_j^t(c)\} \tag{13}$$

where $\gamma_i = \frac{-\frac{1}{M}\sum_c \log P_i^s(c)}{\lambda} - 1$ and $\gamma_j = \frac{-\frac{1}{M}\sum_c \log P_j^t(c)}{\lambda} - 1$. The $\lambda$ is a constant which controls the smoothness of predictions. By imposing $\gamma_i$ and $\gamma_j$, we encourage the smoothness of predictions keeps at a moderate level, which strikes a balance between cross-domain association and data fitting.

Overall, the full objective for the adaptation process is

$$\mathcal{L}^{full} = \mathcal{L}^{ce} + \beta_1 \mathcal{L}^{lov} + \beta_2 \mathcal{L}^{asso} + \beta_3 \mathcal{L}^{lsr} \tag{14}$$

where $\beta_1$, $\beta_2$ and $\beta_3$ are constants controlling the strength of corresponding loss.

**Inference.** At the inference time, we use the adapted network without any association operation. As the aggregated target features are adopted to associate the target pixels with the source ones during training, it is expected that the distribution of aggregated target features is closer to that of source features. Thus we perform the same aggregation as indicated in Eq. (7) at the inference stage.

## 4 Experiments

### 4.1 Dataset

We validate our method on two representative domain adaptive semantic segmentation benchmarks, *i.e.* transferring from the synthetic images (GTAV or SYNTHIA) to the real images (Cityscapes). Both GTAV and SYNTHIA are synthetic datasets. GTAV contains 24,966 images with the resolution around $1914 \times 1024$. SYNTHIA contains 9,400 images with the resolution of $1280 \times 760$. The Cityscapes consists of high quality street scene images captured from various cities. Images are with resolution of $2048 \times 1024$. The dataset is split into a training set with 2,975 images and a validation set with 500 images. For the task GTAV $\rightarrow$ Cityscapes, we report the results on the common 19 classes. For the task SYNTHIA $\rightarrow$ Cityscapes, we follow the conventional protocol [7, 9, 54, 42], reporting the results on 13 and 16 common classes. For all of the tasks, the images in the Cityscapes training set are used to train the adaptation model, and those in the validation set are employed to test the model's performance. We employ the mean Intersection over Union (mIoU) as the evaluation metric which is widely adopted in the semantic segmentation tasks.

### 4.2 Implementation Details

We adopt the DeepLab-V2 [8] architecture with ResNet-101 as the backbone. As a common practice, we adopt the ImageNet [14] pretrained weights to initialize the backbone. For the training, we train our model based on stochastic gradient descent (SGD) with momentum of 0.9 and weight decay of $5 \times 10^{-4}$. We employ the poly learning rate schedule with the initial learning rate at $2.5 \times 10^{-4}$. We train about 11K and 3K iterations with batch size of 8, for GTAV $\rightarrow$ Cityscapes and SYNTHIA $\rightarrow$ Cityscapes respectively. For all of the tasks, we resize images from both domains with the length of shorter edge randomly selected from $[760, 988]$, while keeping the aspect ratio. The $730 \times 730$ images are randomly cropped for the training. Random horizontal flipping is also employed. The resolution for building the associations is at $92 \times 92$. At the test stage, we first resize the image to $1460 \times 730$ as input and then upsample the output to $2048 \times 1024$ for evaluation. In all of our experiments, we set $\beta_1$, $\beta_2$, $\beta_3$ to 0.75, 0.1, 0.01. We set $\lambda$ to 10.0 which is equivalent to restricting the highest predicted probability among classes at around 0.999.

Table 1: Ablation studies based on task GTAV → Cityscapes and SYNTHIA → Cityscapes. The "source-only" means the model trained using source data only, while "source+target" represents the adaptation with labeled source data and unlabeled target data. The $\mathcal{L}^{lov}$, $\mathcal{L}^{fass}$, $\mathcal{L}^{cass}$, and $\mathcal{L}^{lsr}$ in Eq. (14) are progressively added to show their effectiveness. The mIoUs on the Cityscapes test set with respect to 19 and 16 classes are reported for the task GTAV → Cityscapes and SYNTHIA → Cityscapes, respectively. Our full method is denoted as "PLCA". The "Sim-PLCA" means the PLCA is trained by directly encouraging the pixel-wise similarities, while the "PLCA w/o. SAGG" denotes PLCA is trained without the spatial aggregation module discussed in Section 3.3.

| Source Dataset | source-only | | source + target | | | | | |
|---|---|---|---|---|---|---|---|---|
| | $\mathcal{L}^{ce}$ | $+\mathcal{L}^{lov}$ | $+\mathcal{L}^{fass}$ | $+\mathcal{L}^{cass}$ | $+\mathcal{L}^{lsr}$ | PLCA | Sim-PLCA | PLCA w/o. SAGG |
| GTAV | 31.5 | 34.3 | 39.7 | 47.4 | 47.7 | **47.7** | 41.8 | 45.6 |
| SYNTHIA | 35.4 | 36.4 | 40.9 | 46.3 | 46.8 | **46.8** | 42.5 | 44.6 |

Table 2: Comparisons with the state-of-the-arts on task GTAV → Cityscapes. The methods we compared to can be roughly categorized into two groups, *i.e.* "AT" and "ST" which denote the adversarial training based methods and self-training based methods, respectively. All the methods are based on DeepLab-V2 with ResNet-101 as the backbone for a fair comparison.

| | Method | road | side. | build. | wall | fence | pole | light | sign | veg. | terrain | sky | person | rider | car | truck | bus | train | motor | bike | mIoU |
|---|---|---|---|---|---|---|---|---|---|---|---|---|---|---|---|---|---|---|---|---|---|
| | | | | | | | | | | | | | | | | | | | | | **GTAV → Cityscapes** |
| Source Only | – | 34.8 | 14.9 | 53.4 | 15.7 | 21.5 | 29.7 | 35.5 | 18.4 | 81.9 | 13.1 | 70.4 | 62.0 | **34.4** | 62.7 | 21.6 | 10.7 | 0.7 | **34.9** | 35.7 | 34.3 |
| AdaptSeg[41] | AT | 86.5 | 36.0 | 79.9 | 23.4 | 23.3 | 23.9 | 35.2 | 14.8 | 83.4 | 33.3 | 75.6 | 58.5 | 27.6 | 73.7 | 32.5 | 35.4 | 3.9 | 30.1 | 28.1 | 42.4 |
| ADVENT[44] | AT | 89.4 | 33.1 | 81.0 | 26.6 | **26.8** | 27.2 | 33.5 | 24.7 | 83.9 | 36.7 | 78.8 | 58.7 | 30.5 | 84.8 | 38.5 | 44.5 | 1.7 | 31.6 | 32.4 | 45.5 |
| CLAN[31] | AT | 87.0 | 27.1 | 79.6 | 27.3 | 23.3 | 28.3 | 35.5 | 24.2 | 83.6 | 27.4 | 74.2 | 58.6 | 28.0 | 76.2 | 33.1 | 36.7 | 6.7 | 31.9 | 31.4 | 43.2 |
| DISE[7] | AT | 91.5 | 47.5 | **82.5** | 31.3 | 25.6 | 33.0 | 33.7 | 25.8 | 82.7 | 28.8 | **82.7** | 62.4 | 30.8 | 85.2 | 27.7 | 34.5 | 6.4 | 25.2 | 24.4 | 45.4 |
| SSF-DAN [15] | AT | 90.3 | 38.9 | 81.7 | 24.8 | 22.9 | 30.5 | **37.0** | 21.2 | 84.8 | 38.8 | 76.9 | 58.8 | 30.7 | **85.7** | 30.6 | 38.1 | 5.9 | 28.3 | 36.9 | 45.4 |
| PatchAlign [42] | AT | **92.3** | 51.9 | 82.1 | 29.2 | 25.1 | 24.5 | 33.8 | **33.0** | 82.4 | 32.8 | 82.2 | 58.6 | 27.2 | 84.3 | 33.4 | 46.3 | 2.2 | 29.5 | 32.3 | 46.5 |
| MaxSquare[9] | ST | 89.4 | 43.0 | 82.1 | 30.5 | 21.3 | 30.3 | 34.7 | 24.0 | 85.3 | **39.4** | 78.2 | 63.0 | 22.9 | 84.6 | 36.4 | 43.0 | 5.5 | 34.7 | 33.5 | 46.4 |
| CRST[54] | ST | 91.0 | **55.4** | 80.0 | 33.7 | 21.4 | **37.3** | 32.9 | 24.5 | 85.0 | 34.1 | 80.8 | 57.7 | 24.6 | 84.1 | 27.8 | 30.1 | **26.9** | 26.0 | **42.3** | 47.1 |
| ours | – | 84.0 | 30.4 | 82.4 | **35.3** | 24.8 | 32.2 | 36.8 | 24.5 | **85.5** | 37.2 | 78.6 | **66.9** | 32.8 | 85.5 | **40.4** | **48.0** | 8.8 | 29.8 | 41.8 | **47.7** |

## 4.3 Ablation Studies

Table 1 demonstrates the improvement of mIoU compared to the "source-only" baseline, by progressively adding each proposed module into the system. It can be seen that each module contributes to the final success of the adaptation. Finally, we achieve 47.7% and 46.8% mIoU with GTAV and SYNTHIA as the source respectively, outperforming the "source-only" by +13.4% and +10.4%.

**Effect of multi-level associations.** As shown in Table 1, additionally performing the association on predictions (*i.e.* imposing $\mathcal{L}^{cass}$) brings obvious improvement, implying the multi-level associations complement each other to mine the similarities across domains and mitigate the domain shift.

**Effect of contrastively strengthening the associations.** As discussed in Section 3.2, for the associated pairs of source and target pixels, we can strengthen their connections by directly encouraging their feature similarities. In Table 1, we compare PLCA (which penalizes $\mathcal{L}^{asso}$) to "sim-PLCA" (which directly encourages the feature similarities). It can be seen that PLCA performs significantly better than "Sim-PLCA", verifying the effectiveness of contrastively strengthening the associations.

**Effect of spatial aggregation on target.** We remove the spatial aggregation module discussed in Section 3.3 to verify its effect and necessity. As shown Table 1, as expected, without adopting the spatial aggregation on target, the mIoU of the adapted model decreases noticeably, implying the necessity of reducing the domain discrepancy for more diverse target pixels.

## 4.4 Comparisons with the state-of-the-arts

We compare our method with previous state-of-the-arts in Table 2 and Table 3. For previous methods, we directly cite the published results from corresponding papers for a fair comparison. All the methods we compared to are based on DeepLab-V2 with ResNet-101 as the backbone, which is the same with our method. Roughly speaking, previous methods can be categorized into two groups:

Table 3: Comparisons with the state-of-the-arts on task SYNTHIA → Cityscapes. We report the mIoUs with respect to 13 classes (excluding the "wall", "fence", and "pole") and 16 classes. All the methods are based on DeepLab-V2 with ResNet-101 as the backbone for a fair comparison.

| | Method | road | side. | build. | wall* | fence* | pole* | light | sign | veg. | sky | person | rider | car | bus | motor | bike | mIoU | mIoU* |
|---|---|---|---|---|---|---|---|---|---|---|---|---|---|---|---|---|---|---|---|
| | | | | | | **SYNTHIA → Cityscapes** | | | | | | | | | | | | | |
| Source Only | – | 51.2 | 21.8 | 67.8 | 8.2 | 0.1 | 26.2 | 17.7 | 17.3 | 69.2 | 67.1 | 52.7 | 22.8 | 62.3 | 31.6 | 21.0 | 46.1 | 36.4 | 42.2 |
| AdaptSeg[41] | AT | 84.3 | 42.7 | 77.5 | – | – | – | 4.7 | 7.0 | 77.9 | 82.5 | 54.3 | 21.0 | 72.3 | 32.2 | 18.9 | 32.3 | – | 46.7 |
| CLAN[31] | AT | 81.3 | 37.0 | 80.1 | – | – | – | 16.1 | 13.7 | 78.2 | 81.5 | 53.4 | 21.2 | 73.0 | 32.9 | 22.6 | 30.7 | – | 47.8 |
| SSF-DAN[15] | AT | 84.6 | 41.7 | 80.8 | – | – | – | 11.5 | 14.7 | 80.8 | **85.3** | 57.5 | 21.6 | 82.0 | 36.0 | 19.3 | 34.5 | – | 50.0 |
| ADVENT[44] | AT | 85.6 | 42.2 | 79.7 | 8.7 | 0.4 | 25.9 | 5.4 | 8.1 | 80.4 | 84.1 | 57.9 | 23.8 | 73.3 | 36.4 | 14.2 | 33.0 | 41.2 | 48.0 |
| DISE [7] | AT | **91.7** | **53.5** | 77.1 | 2.5 | 0.2 | 27.1 | 6.2 | 7.6 | 78.4 | 81.2 | 55.8 | 19.2 | 82.3 | 30.3 | 17.1 | 34.3 | 41.5 | 48.7 |
| PatchAlign [42] | AT | 82.4 | 38.0 | 78.6 | 8.7 | 0.6 | 26.0 | 3.9 | 11.1 | 75.5 | 84.6 | 53.5 | 21.6 | 71.4 | 32.6 | 19.3 | 31.7 | 40.0 | 46.5 |
| MaxSquare[9] | ST | 82.9 | 40.7 | 80.3 | 10.2 | 0.8 | 25.8 | 12.8 | 18.2 | 82.5 | 82.2 | 53.1 | 18.0 | 79.0 | 31.4 | 10.4 | 35.6 | 41.4 | 48.2 |
| CRST [54] | ST | 67.7 | 32.2 | 73.9 | 10.7 | **1.6** | **37.4** | 22.2 | **31.2** | 80.8 | 80.5 | 60.8 | **29.1** | 82.8 | 25.0 | 19.4 | 45.3 | 43.8 | 50.1 |
| ours | – | 82.6 | 29.0 | **81.0** | **11.2** | 0.2 | 33.6 | **24.9** | 18.3 | **82.8** | 82.3 | **62.1** | 26.5 | **85.6** | **48.9** | **26.8** | **52.2** | **46.8** | **54.0** |

1) adversarial training based methods (denoted as "AT"); 2) self-training based methods (denoted as "ST"). It can be seen our method outperforms previous state-of-the-art adversarial training based methods, and performs comparable to or event better than the self-training based methods.

Specifically, for the task GTAV → Cityscapes and task SYNTHIA → Cityscapes, our method outperforms the adversarial training based method "AdaptSeg" by +5.3% and +7.3% respectively, and outperforms "SSF_DAN" by +2.3% and +4.0% respectively. The results verify that our way of taking the pixel-wise relationships into account benefits the adaptation.

From Table 2 and Table 3, it can be seen that our method also performs favorably against previous state-of-the-art self-training based methods. For the task GTAV → Cityscapes, we achieve 47.7% mIoU, which outperforms CRST (47.1%) by 0.6%. While for the task SYNTHIA → Cityscapes, our method outperforms CRST by more than 3%. For the self-training based methods, we notice pyCDA [27] achieves 47.4% and 46.7% mIoUs on task GTAV → Cityscapes and SYNTHIA → Cityscapes respectively. However, it is based on the PSPNet architecture [50], which has been shown [50, 10] to perform better than DeepLab-V2. Thus it is not listed in the tables. Despite this, we achieve slightly better results compared to it. Moreover, our method is orthogonal to the self-training methods, and can be combined with self-training to improve the performance further.

Overall, the results shown in Table 2 and Table 3 illustrate our method performs favorably against previous state-of-the-arts, verifying the effectiveness of our method.

### 4.5 Visualization

We visualize the masks predicted by our PLCA and compare our results to those predicted by the "source-only" network in Fig. 3. The masks predicted by PLCA are smoother and contain less spurious areas than those predicted by the "source-only" model, showing that with PLCA, the pixel-level accuracy of predictions has been largely improved.

## 5 Conclusion

In this paper, we propose to exploit the abundant cross-domain pixel-wise similarities to mitigate the domain shift, via cyclically associating cross-domain pixels, which provides a new perspective to deal with the domain adaptive semantic segmentation. Our method can be trained end-to-end in one stage, achieving superior performance to previous state-of-the-arts. In future, we will try to apply associations to multiple layers of features and evaluate our method on more recent deep architectures. We hope our exploration can provide a solid baseline and inspires future research in this area.

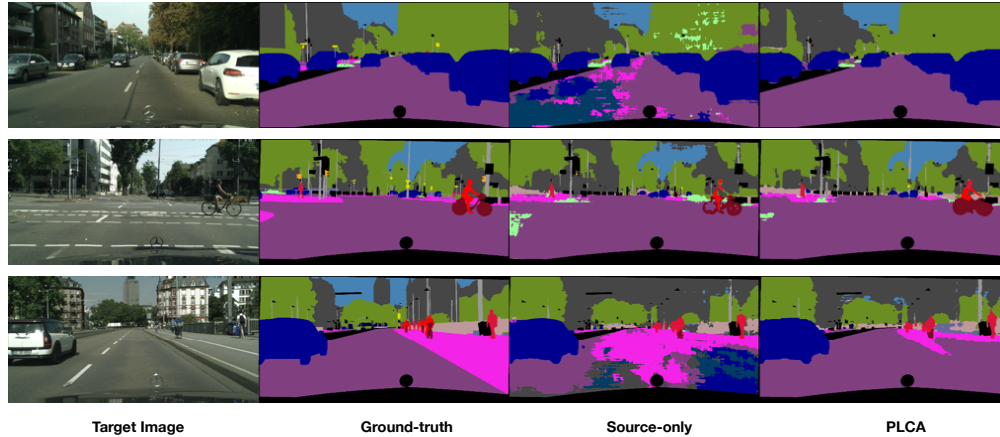

| Target Image | Ground-truth | Source-only | PLCA |

Figure 3: Visualization of predicted segmentation masks on the test images of Cityscapes. From the left to the right, the original image, the ground-truth segmentation mask, the mask predicted by "source-only" network, and the mask predicted by our PLCA are shown one by one. Both of the "source-only" network and PLCA are trained with GTAV as the source.

## Broader Impact

Our research may benefit the people or communities who want to apply semantic segmentation in various scenarios but have no annotations. It will reduce the cost to annotate the out-of-domain data, which is economic and friendly to the environment. Our method makes the trained segmentation system more robust when the system is deployed in a distinct scenario. Thus, for some applications, it will reduce the risk of accident. Our method does not leverage the bias in the data. However, if the collected training data is biased, the adapted system may be affected more or less.

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
