[Supplementary Material]

# Supplementary Material
# Pixel-Level Cycle Association: A New Perspective for Domain Adaptive Semantic Segmentation

**Guoliang Kang**[1]**, Yunchao Wei**[2]**, Yi Yang**[2]**, Yueting Zhuang**[3]**, Alexander G. Hauptmann**[1]
[1] School of Computer Science, Carnegie Mellon University
[2] ReLER, University of Technology Sydney [3] Zhejiang University
kgl.prml@gmail.com, alex@cs.cmu.edu
{yunchao.wei, yi.yang}@uts.edu.au
yzhuang@zju.edu.cn

## Sensitivity Analysis of Hyper-parameters

We investigate the sensitivity of our method to the hyper-parameters $\beta_1$, $\beta_2$ and $\beta_3$. The experiments are conducted on the task SYNTHIA $\rightarrow$ Cityscapes for an example. The trend on another task GTAV $\rightarrow$ Cityscapes is similar. As shown in Fig. A1, on the whole, our method is not sensitive to those hyper-parameter settings. In a vast range, our method outperforms previous state-of-the-art method obviously on this task. The bell curve shape of $\beta_2$ implies the regularization effect of proposed association loss $\mathcal{L}^{asso}$, and we need to strike a balance between fitting to the source and minimizing the domain discrepancy to achieve the optimal results.

Figure A1: Sensitivity analysis of hyper-parameters $\beta_1$, $\beta_2$ and $\beta_3$. Our method (PLCA) and previous state-of-the-art adversarial training based method (SOTA) are denoted as the red and blue curve respectively. The experiments are conducted on the task SYNTHIA $\rightarrow$ Cityscapes for an example and the mIoU with respect to 16 classes is reported. The trend on another task GTAV $\rightarrow$ Cityscapes is similar.

## Does Multi-Layer Association Help?

In our paper, we apply the pixel-level cycle association to the last layer feature maps of the backbone and the final network outputs. Technically, pixel-level cycle association can also be applied to other layers of features in deep CNNs. However, empirically we find that it brings minor improvement, but introduces much additional computation cost. We remain it as an open question to further investigate in future.

## Correct Ratio of Cycle-Consistent Associations

Among all the cycle-consistent associations, the average correct ratio is above 70% during training. In terms of the cycle association (Sec. 3.2), rarely-presented classes are relatively harder to match (around 10% can be associated). However, our proposed diffusion module (Sec. 3.3) encourages a diverse set of pixels to be adapted, and enables the adaptation of rarely-presented classes via other similar associated classes. Empirically we find that our method works better on the rarely-presented classes, than the source-only baseline and previous state-of-the-art. For example, on the task SYNTHIA → Cityscapes, our method achieves 35.9% mean IoU over the classes with rarest pixels (*i.e.* motorcycle, rider, traffic light, bus, and bicycle), outperforming the source-only baseline (27.8%) and previous state-of-the-art (28.2%) by a large margin.