[Reviews · NeurIPS 2020]

Review 1

Summary and Contributions: The authors propose a novel method for domain adaptation of semantic segmentation networks in the challenging synthetic to real scenario. The work mainly proposes to align the feature representation among the two domains along parts of the images which share the same visual context. These parts can be automatically identified by deep feature correlations and cycle consistency. Given a couple of images, a source deep feature is spatially matched to the most similar target one, tha same process is repeated for the selected target feature and all the source ones. If the final matching source feature shares the same semantic label as the initial source feature, the cycle is considered a success and the triplet of features can be used to force a similarity among features extracted across domains. This technique together with other regularization allows this method to achieve state of the art results in the two considered datasets.

Strengths: + I really liked the idea of matching deep features by deep correlation to automatically identify portions of the images with the same visual appearance. I find it quite novel and at the same time it is quite intuitive why we would want to make the representation of only subsect of the image features similar instead of aligning everything. + My intuition on the method, and a slightly different point of view from the one presented on the paper, is that the proposed gradient diffusion by spatial aggregation basically provides an elegant and self-supervised way of performing unsupervised image segmentation (see Fig. 2). Then by matching this feature with two coherent ones from the source image the method is implicitly doing some sort of soft-labeling of the target sample. + The whole system can be trained end-to-end without requiring cumbersome process or subsequent training stages plus it seems to be pretty robust with respect to the choice of the hyperparameters as discussed in the supplementary material. + The ablation study in Tab. 2 is concise but provides all the useful information to show the effectiveness of the proposed loss functions and regularizations.

Weaknesses: - The method has been tested on a single architecture even if, as the authors states at line 243, new architecture like PSPNet might perform even better. - Some more recent works that might be highly correlated needs to be cited and compared against, for example this work “Differential Treatment for Stuff and Things: A Simple Unsupervised Domain Adaptation Method for Semantic Segmentation - Wang et al. cvpr2020” might share some similarity with the proposed method that need to be discussed in context. - The method has been tested only among synthetic and real datasets that share quite a lot of similarities. It’s a little bit unclear how well the proposed feature matching strategy generalizes to datasets with bigger domain discrepancies. - Some implementation details can be made more clear in the paper

Correctness: I believe the claim to be correct and the experimental evaluation methodology to be correct.

Clarity: The paper is well written and easy to follow. Some minor implementation details can be made more clear, but this can be addressed in a polishing of the paper before camera ready.

Relation to Prior Work: Previous works have mostly been properly cited and discussed in this manuscript. However some more recent and highly related works should be added and discussed, some of them outperform the proposal but I still think that this work it’s pretty valuable. For example: a. Bidirectional Learning for Domain Adaptation of Semantic Segmentation - Li et al. 2019 b. Learning Texture Invariant Representation for Domain Adaptation of Semantic Segmentation - Kim et al. 2020 c. Differential Treatment for Stuff and Things: A Simple Unsupervised Domain Adaptation Method for Semantic Segmentation - Wang et al. 2020 d. Unsupervised Intra-domain Adaptation for Semantic Segmentation through Self-Supervision - Pan et al. 2020

Reproducibility: Yes

Additional Feedback: I am giving an 8 because I considered this submission very interesting and quite novel. I have some doubts that I would like the authors to clarify in the rebuttal and, eventually, in a revised version of the paper: (a) Is there a reason why the method has not been tested with newer architectures than DeepLab v2, e.g. the cited PSPNet? (b) How do you select which source pixels to use to define pixel level associations? Are all source pixels of a certain image considered as candidates? (c) At which resolution is the pixel level association between source and target pixel computed? (d) You have mentioned using your method together with a self-training method but how would you carry out the integration? Simply pseudo-labeling target samples and adding an additional loss on it? Or would you integrate the pseudo-labeling coming from self-training into the feature matching and cycle consistency formulation? ===== Post rebuttal comments ===== The authors have correctly addressed my few concerns on the rebuttal, therefore I'm keeeping my original rating and i would sugges accepting this work to neurips


Review 2

Summary and Contributions: This paper proposes a domain adaptation technique by finding cycle-consistent pixels between source and target images and reinforcing. That is, given a pixel in the source, finding the nearest neighbor in a target, then finding that point’s nearest neighbor back in the source, and checking that the original and final source points belong in the same class. If so, they are brought together in feature space, away from the other points (in a contrastive manner). This is different than cycada (which aims to stylize the pixels) or feature discrepancy minimization methods.

Strengths: The paper proposes to mine for cycle-consistent pixel associations: going from source to target back to source, using nearest neighbors, and checking if the start and end points are in the same class. These points are then brought together in feature space, in contrastive to other points in the image. To my knowledge, this method is novel and different than previous methods. The paper explores this idea quite thoroughly. The method performs this in feature space (Sec 3.2) and output space (3.4), with some feature diffusion such that a more diverse set of points are selected (Sec 3.3), and with contrast normalization on features. The paper validates each of these design decisions (along with contrastive vs “simply” associating cycle-consistent points together) in Table 2. The paper beats state-of-the-art methods.

Weaknesses: Section 3.2 proposes associating pixels through cycle-consistency in a feature representation. One dimension the paper does not study (that I see) is the receptive field, or how deep in the network these features are extracted from. Is the method sensitive to this design choice? Can the system benefit from perhaps performing this operation in every (or multiple) layers of a feature extractor, such as in VGG perceptual loss? I also believe the mechanism of finding nearest neighbors could be better studied and visualized. Figure 2 does visualize qualitative similarity maps. How often do the cycle-consistent associations find a associated pixel in the target domain of the same, correct label? Is it difficult to find matches with classes that are not not often represented?

Correctness: Yes, the paper’s claims and methodology seem correct.

Clarity: I was able to understand the paper and method.

Relation to Prior Work: While the related work covers domain adaptation literature, I believe it can draw some further connections outside of this immediate area. For example, [1] learns features guided by cycle consistency and [2] finds correspondences between images by cycle-consistent feature matching for graphics applications. Furthermore, the work uses the contrastive loss, which has seen popular use through the unsupervised learning community [3] and shown benefits in knowledge distillation [4]. [1] Zhou et al. Learning dense correspondence via 3d-guided cycle consistency. CVPR 2016. [2] Aberman et al. Neural Best-Buddies: Sparse Cross-Domain Correspondence. SIGGRAPH 2018. [3] van den Oord et al. Contrastive Predictive Coding. 2018. [4] Tian et al. Contrastive Representation Distillation. ICLR 2020.

Reproducibility: Yes

Additional Feedback: Overall, I was able to understand the method. To my knowledge, it is novel and different than previous methods. The paper studied various aspects and design decisions related to this idea through an ablation study. The method also outperforms previous methods. I do think studying the feature extractor to greater detail would make the paper stronger, along with more connections to previous literature. Overall, I believe this is a solid submission. ----------- I thank the authors for the additional information in the rebuttal. I hope they can be incorporated in an updated revision or supplementary material.


Review 3

Summary and Contributions: This paper addresses the problem of unsupervised domain adaptation for semantic segmentation, where the source data with annotated labels and the target data without labels are available. It proposes to enhance the similarities between cycle-consistent pixels between source and target images, compared to other pixel-pairs. To address the problem that the cycle-consistent pixels are sparse, a spatial aggregation module is used so as to back-propogation gradients across all pixels in the training images. Experiments on two adaptation cases show the effectiveness of the proposed method.

Strengths: The proposed method is technically sound and reasonable. While existing methods for domain adaptation are to minimize the distribution discrepancy, this paper proposes a new perspective for domain adaptation, which finds and minimizes the possible pixel-pairs belonging to the same category across domains. The basic idea is reasonable and somewhat straightforward, but the paper find a good way to incorporate this idea into addressing the domain adaptation problem end-to-end. The paper has strong experimental results. The ablation study is also helpful to understand the method.

Weaknesses: The loss contains many parts, which makes tuning the weights for these parts could be a tedious task. The paper lacks studies on how to tune these parameters, and how they will influence the final results. It is unclear whether the proposed method is sensitive to them or not.

Correctness: It is technically sound.

Clarity: Yes, the paper is well organized and easy to follow.

Relation to Prior Work: The relation to previous works could be improved by describing more about existing works and how the proposed one differs to them. Currently, it only very briefly describes three main categories of domain adaptation methods.

Reproducibility: Yes

Additional Feedback: After rebuttal, I maintain my initial recommendataion.

[Author Response · NeurIPS 2020]

Thanks for all the reviewers' positive feedback and constructive suggestions. We will revise our paper accordingly.

**To Reviewer 1**

**Q1:** Reason why the method hasn't been tested with newer architectures than DeepLab v2.

**A1:** Due to the historical reasons, most previous methods are tested with DeepLab v2. To facilitate a fair comparison,
we chose DeepLab v2 as our base architecture. And we will supplement more results based on recent state-of-the-art
architectures in our revised version, to setup new baselines for future research.

**Q2:** The selection of source pixels to define pixel level association. Are you using all as candidates?

**A2:** Yes, we used all the source pixels of a certain image as candidates, other than the ones out of the label set. We will
revise this part and make it clearer.

**Q3:** At which resolution is the pixel level association between source and target pixel computed?

**A3:** The resolution for building the associations is at $92 \times 92$, which is around $1/8$ times as the input resolution $730 \times$
$730$. We will add this detail in our revised manuscript.

**Q4:** The way to integrate your method together with a self-training method.

**A4:** There are multiple ways to integrate our method with the self-training. The most conventional way is to use the
self-training as a second stage training, after achieving the adaptation model using our method. Another possible way,
as discussed by the reviewer, is to integrate the pseudo-labeling into the feature matching. Specifically, an additional
cycle-association loss, based on the association starting and ending at the *target* pixel, could be imposed. We remain
this as a future work.

**Q5:** More related work and more implementation details.

**A5:** Thanks. We will cite the suggested papers and add more implementation details in our revised manuscript.

**To Reviewer 2**

**Q1:** How deep in a network the features should be extracted from. Is multi-layer feature association helpful?

**A1:** 1) For the feature extractor, as we discussed in Line 145, we applied pixel-level cycle association to the last-layer
feature maps of the backbone. And through back-propagation, the features in previous layers can also be adapted. 2)
We have attempted to impose the cycle association loss on the multiple layers of feature maps. Empirically we found
that it brings minor improvement, but introduces much additional computation cost. And as most previous works didn't
align multi-layer feature maps, we chose to only associate the last-layer features to facilitate a relatively fair comparison.
3) However, we believe it remains an open question and valuable to further investigate in future.

**Q2:** How often do the cycle-consistent associations find a associated pixel in the target domain of the same, correct
label? Is it difficult to find matches with classes that are not often represented?

**A2:** 1) Among all the cycle-consistent associations, the average correct ratio is above 70% during training. 2) In
terms of the cycle association (Sec. 3.2), rarely-presented classes are relatively harder to match (around 10% can be
associated). However, our proposed diffusion module (Sec. 3.3) encourages a diverse set of pixels to be adapted, and
enables the adaptation of rarely-presented classes via other similar associated classes. Empirically we found that our
method works better on the rarely-presented classes, than the source-only baseline and previous state-of-the-art. For
example, on the task SYNTHIA $\rightarrow$ Cityscapes, our method achieves 31.7% mean IoU over the classes with rarest pixels
(*i.e.* motocycle, rider, traffic light, bus, and bicycle), outperforming the source-only baseline (24.0%) and previous
state-of-the-art (28.2%) by a large margin.

**Q3:** More connections to previous works which are out of the domain adaption area.

**A3:** Thanks. We will cite the suggested related papers and add more discussions about the connections to them.

**To Reviewer 4**

**Q1:** The sensitivity of the model to the hyper-parameter settings.

**A1:** We have discussed the selection of $\alpha$ in Line 134-135. And we have shown the sensitivity of our model to the other
hyper-parameters $\beta_1$, $\beta_2$, $\beta_3$ in the supplementary material due to the space limit. We found that our method is robust to
the choices of hyper-parameters and achieves consistent superior performance compared to previous state-of-the-arts.

**Q2:** More discussions about previous work.

**A2:** Thanks. We will add more discussions about how our method differs from existing works in our revision.

[Meta-Review · NeurIPS 2020]

All reviewers very much enjoyed this paper and recommended acceptance. Reviewers appreciated the novelty of the method, the thoroughness of the experiments, and the quality of the results. The rebuttal clarifies several details that should be added to the paper, and promises results on more recent architectures, which will be great to see.